# The Influence of B_4_C Film Density on Damage Threshold Based on Monte Carlo Method for X-ray Mirror

**DOI:** 10.3390/ma17051026

**Published:** 2024-02-23

**Authors:** Tingting Sui, Haohui Zhuo, Anchun Tang, Xin Ju

**Affiliations:** Department of Physics, School of Mathematics and Physics, University of Science and Technology Beijing, Beijing 100083, China; aa953090814@163.com (H.Z.); b20200357@xs.ustb.edu.cn (A.T.)

**Keywords:** B_4_C film, density, XFEL, Monte Carlo, damage threshold

## Abstract

The uniformity and consistency of X-ray mirror film materials prepared by experimental methods are difficult to guarantee completely. These factors directly affect the service life of free electron laser devices in addition to its own optical properties. Therefore, the quality of the film material, especially the density, has a critical effect on its application. Boron carbide film and monocrystalline silicon substrate were suitable examples to explore their influence of density on the damage threshold based on Monte Carlo and heat-conduction methods. Through simulation results, it was found that the change in film density could affect the energy deposition depth and damage threshold. When the film density was 2.48 g/cm^3^, it had relatively high damage threshold in all energy ranges. And then the specific incident parameter for practical application was investigated. It was found that the damage mechanism of the B_4_C/Si was the melting of the interface. And the damage threshold was also higher with the film density of 2.48 g/cm^3^. Therefore, it was recommended to maintain the density at this value as far as possible when preparing the film, and to ensure the uniformity and consistency of the film material.

## 1. Introduction

The development and progress of light sources promote the further improvement of science and technology. These developments can help humans understand and change the unknown world. In particular, in the 1960s, the invention of the laser promoted great advances in the scientific frontier, greatly improving the time and spatial resolution. However, due to the limitation of amplification medium, it was difficult to realize the application of vacuum ultraviolet and X-ray bands. At the beginning of the 21st century, development of the first free electron laser amplification experiment was achieved at Stanford University in the United States, opening a new era of X-ray light sources [1,2].

The X-ray free electron laser (XFEL) consists of strongly coherent pulsed X-rays which are produced by a relativistic electron beam. It utilizes self-amplified and self-excited radiation under the periodic magnetic field [3,4,5]. More importantly, it could provide the possibility of studying matter at atomic space scales and femtosecond timescales due to the properties of ultra-short pulse width, ultra-fast time structure and ultra-high photon flux [6,7,8,9,10,11]. The beamline system is a very important part of XFEL devices, which is the bridge connecting the light source and the experimental station. It can efficiently transmit the coherent X-ray laser emitted by the accelerator to the experimental end. X-ray mirror is the crucial optical component of XFEL system to realize beam deflection, splitting and focusing. However, the high characteristics of photon energy and peak power make the X-ray mirror face serious damage and sustainable development problems [12]. This is an important scientific and technological problem that must be faced and solved.

The substrate materials of monocrystalline silicon coating with low-Z materials, such as boron carbide (B_4_C) and silicon carbide (SiC), are mostly used as X-ray mirror in the international light source. In addition, the metals with high Z are also considered as the coating materials, such as rhodium and ruthenium [13,14]. Under the irradiation of high-energy X-ray, the mirror materials will be easily damaged. However, the damage properties of components greatly change with different photon energy levels. For example, in the waveband of ultraviolet and soft X-ray, photons mainly interact with electrons in the outer layer of atoms. It could cause the break of chemical bonds, radiation modification and ablation. And finally, macroscopic damage occurs [15]. Hard X-rays can interact with the inner layer electrons of the atoms, leading to severe damage. Milov et al. researched the damage of Ru films irradiated by femtosecond XUV free-electron laser pulses by experiment and simulations [16]. They found that the damage started from the melting of thin film materials, and the energy transfer progress was complex. In addition, B_4_C-coated bilayer materials under various irradiation conditions were studied by Follath et al. They found that the damage threshold of normal incidence was ~10–100 times smaller than that of grazing incidence [17]. Moreover, various studies were performed to study the damage threshold with the energy below 20 keV, promoting the application for mirror optics of XFELs [14,18].

However, for the same material, the damage performance also different in various states and parameter conditions. For example, boron carbide (B_4_C) possesses the advantages of a high melting point (2450 °C), low density (2.52 g/cm^3^) and high damage threshold [14,19,20,21]. Additionally, it can increase the attenuation length by spreading out any absorbed energy over a larger volume [14]. It is applied to the mirror under the waveband of X-ray and extreme ultraviolet. At present, boron carbide film is mainly prepared by the magnetron sputtering method to obtain the smooth and dense films [22,23]. However, the thickness and surface roughness cannot keep consistent, and the composition and mass density of films will be unstable depending on the process conditions during magnetron sputtering. For example, Li et al. researched the optical properties of boron carbide thin films with different B/C molar ratios [24]. The results showed that different molar ratios could lead to different density and optical properties of B_4_C. Additionally, Song et al. researched the appropriate surface roughness and specific impact on dielectric film properties [25]. However, the film density will also change due to the unstable preparation process. It was also found that the density of B_4_C films varied in different experiments [26,27,28,29,30]. The influence of the density change under irradiation with high energy X-rays is unclear, especially for the damage threshold. It could affect the application for the X-ray mirror. Due to the limitations of the experiment, it is difficult to observe the ultra-fast process and damage effect in situ. Therefore, the relevant simulation method will be important, such as the Monte Carlo method and heat transfer analysis. Attention should be paid to the impact of physical parameter changes on the damage threshold by analyzing the physical process. Moreover, it could simulate the damage threshold with the specific incident parameter designed for practical application. It is of great significance to guide the preparation and application for XFEL mirror films.

Therefore, with the simulation method, the influence of the material density on the damage threshold under specific input parameters will be investigated. In this work, we take boron carbide (B_4_C) as the object to study the influence of its density parameters on the damage threshold. The energy deposition depth and damage threshold with different photon energy levels at various densities of B_4_C film will be investigated firstly. And then, in order to improve the damage threshold with high incident energy, the designed grazing incident angle for practical application will be applied. More importantly, the damage mechanism of the B_4_C/Si will be researched. In addition, the optimum film density to ensure the damage threshold will be explored under the specific incident parameters. It will provide guidance for the preparation of mirror film materials and process optimization, and could also promote the damage threshold study of X-ray mirror film.

## 2. Computational Method and Details

### 2.1. Computational Method

In this work, non-equilibrium electron kinetics progress was performed to describe the interaction between photons and materials by Monte Carlo method with Geant4 11.0 software [31,32,33]. Monte Carlo is a stochastic simulation and computational method based on probability and statistical theory, which uses random numbers to solve the dynamic process of system evolution. One of its important applications is the simulation of particle transport and collision processes in matter. It could be used to mathematically solve the Boltzmann equation [34]. The basic assumption of the particle transport problem was that there was no interaction, polarization and external forces. Moreover, the number was enough to ignore deviations caused by statistical fluctuations. The basic procedure of the Monte Carlo simulation method was to treat the particle into a classical state whose step size was determined by the mean free path or the total scattering cross section, and each scatting was regarded as another random step. In addition, the change in the angle and energy during scattering was also determined by the corresponding scattering cross section. Geant4 is an open-source software based on Monte Carlo method to simulate the transport progress of particles in materials. C ++ was used as the basic language for Geant4 software developed by European Organization for Nuclear Research (CERN) in 1994 [31]. It could be input by the particle source and material related terms, including the type, energy, and position. Properties such as particle-material interactions were determined by Monte Carlo simulation, the probability of which was derived from the database of Geant4 composed by experimental measurements and theoretical calculations. 

In addition, the non-equilibrium thermal effects and thermal diffusion processes between electrons and the lattice were analyzed by heat-conduction methods. This could describe the temperature distribution change with time and space by solving the heat equation for the uneven material. And it was carried out by the enthalpy method [35,36]. It used enthalpy as the dependent variable to solve the heat transfer differential equation. When the material was irradiated by laser, it was easy to carry phase transition due to the large amount of heat in a very short time. Thus, it could describe the heat transfer and accumulation.

### 2.2. Simulation Details

The Monte Carlo simulation of this work is mainly divided into geometric and physical modeling. The geometric diagram is shown in Figure 1. The thickness of the B_4_C film was 50 nm and depth direction of the material was Z. The physical modeling included the light source and the process of interaction with materials. The spatial distribution of the photons was Gaussian distribution with the number of 10^5^. And the area of the irradiation of B_4_C was 7.27 × 10^−7^~1.07 × 10^−6^ m^2^ in order to be consistent with the practical application value. The photoelectric effect, Auger effect and secondary electron cascade would occur according to the cross-section data. The databases used in this study are EPDL97, EPICS2014, EEDL and EADL [37,38,39]. All photoelectrons and secondary electrons were tracked until the threshold energy of thermalized electrons reached below 10 eV. The density of the materials referred to the regular value at room temperature. In the original state, the density of bulk B_4_C is 2.52 g/cm^3^. Due to the limitation of the input parameters, the value range of material density was set to 2.40–2.55 g/cm^3^ [22]. It could represent the change in the material density when the film was prepared. The information such as energy deposition, location and relaxation time were counted.

Since the longitudinal gradient of the temperature was much larger than the transverse gradient, it could be simplified using a one-dimensional heat transfer model. Thus, the process of heat transfer was described by solving the enthalpy change. The formula is as follows [40,41,42]:(1)$$\frac{\partial h}{\partial t}=\frac{\partial }{\partial z}(\frac{k}{C}\frac{\partial h}{\partial z})+S$$
where, *h* is the enthalpy of the materials, and *S*, *z*, *k*, *C* represent the heat source, depth, heat conductivity coefficient and heat capacity, respectively. When the initial enthalpy is 0 in the room temperature, the modified heat source term is as follows [42]:(2)$$S(z,t)=\sqrt{\frac{4\ln2}{\pi }}\frac{F(1-R)(1-\alpha )}{\tau_{p}}\exp(-4\ln2{\left(\frac{t}{\tau_{p}}\right)}^{2})AEF(Z)\sin\theta$$
where, *F*, *R*, *τ_p_*, *θ*, *α* and *AEF*(*Z*) is incident flux, reflectivity, pulse width, grazing incidence angle, photoelectron emissivity and energy absorption, respectively. In addition, α and *AEF*(*Z*) is the data from Monte Carlo.

Here, the damage threshold of the reflective mirror was roughly estimated by the melting of the material. The calculation formula is as follows [42,43]:(3)$$F_{th}=\frac{D_{th}\rho N_{A}d}{A(1-R)\sin\theta }$$
where, *A*, *R*, *θ*, *ρ*, *N_A_*, *d* represent the atomic weight, the reflectivity, the incident angle, the density, the Avogadro constant and the energy deposition depth. *D_th_* = 3*k_B_T* is the melting dose per atom, and k_B_ is the Boltzmann constant.

## 3. Results and Discussion

### 3.1. The Physical Processes and Simulation Accuracy

Generally, there are three main interactions between photons and matter, including the photoelectric effect, Compton scattering and pair production. The photoelectric effect is dominant for low-energy photons and substances with a high atomic number. The photon interacts with the target atom of the material, transferring all energy to one of the bound electrons. The outgoing electrons become photoelectrons and the photons themselves disappear. The photoelectric cross section is related to the atomic number Z and the photon energy. And the Compton effect predominates for substances with an intermediate number of energy photons and a low atomic number. Compton scattering refers to the scattering the incident photon with the orbital electron, transferring some energy to the electron and ejecting it from the atom to become a recoiled electron. Meanwhile, the incident photon loses energy and changes direction to become a scattering photon. The necessary condition for pair production is that the photon incident energy is greater than 1.02 MeV, which is not involved in this work. When a photon beam interacts with matter, no matter what kind of effect occurs, the incident photon will disappear or be converted into a photon with different energy and direction. The intensity of the photon beam will be weakened and attenuated correspondingly. And it follows the Lambert-Beer law. Photons generate secondary electrons through the above forms of interaction. By tracing the secondary electron states, the deposited state of energy could be obtained, which causes heat transfer from the electron to the lattice. Finally, the damage will occur to the materials. All of the interactions are related to the cross section. In this work, the relationship between cross section and photon energy of B_4_C in database was plotted in Figure 2. Accordingly, the cross section derived from the simulation results was shown in Table 1, compared with that of database [44]. The results showed that the cross section derived from the simulation was in accordance with that in the database. It could prove the correctness of the physical progress. When the incident photon energy was 3 keV, the cross section of photoelectric absorption was far greater than that of other progress. It could mean that almost only photoelectric absorption occurred in this system. Additionally, the probability of coherent and incoherent scattering was equal in photon energy with 10 keV. In addition, it needed to emphasize that the coherent scattering was referred to elastic scattering—only the photon propagation direction changed and energy was unchanged. Accordingly, the incoherent scattering changed both the energy and direction of the photon. 

In addition, to guarantee the correct of the physical progress, the accuracy of the model was also important. Therefore, the reflectivity and damage threshold with the bulk density of B_4_C (2.52 g/cm^3^) was studied. The angle of incidence was 2 mrad. And the incident energy was selected as 3, 10, 15, 20, 25 keV. The reflectivity was calculated by the published references [45]. And the results of the damage threshold and film reflectivity varying with photon energy are shown in Figure 3. The results showed that the damage threshold was gradually increased with the photon energy before 10 keV at a certain incident angle with 2 mrad. It was mainly because that the photoelectric absorption cross section was one order of magnitude larger than the scattering cross section. The photoelectric absorption mainly occurred. Thus, the photoelectron kinetic energy was related to the photon energy, resulting in the deeper deposition depth. The damage threshold accordingly increased. When the photon energy exceeded 10 keV, the damage threshold slightly dropped, mainly due to the slight decrease of reflectivity from 0.9989 to 0.9974. However, when the photon energy was above 15 keV, the damage threshold had a significant decline due to the decrease of the reflectivity. It could mean that when the incident photon energy exceeded 15 keV, the incident angle should be less than 2 mrad for applications of B_4_C/Si mirror systems [37,38,39]. The simulation result was consistent with that of published work [40]. It confirms that the model was corrected.

### 3.2. Influence of B_4_C Density Change on Damage Threshold at Grazing Incident Angle of 2 Mrad

However, in Monte Carlo simulation, the material itself was set by density parameters and atomic number, coupled with the material database. It was difficult to directly describe the details of the geometric model. It could be assumed that there was inhomogeneity in the preparation and processing, such as defects or dislocation, which would affect the density of the film. It would lead to a decrease in density [46]. The density generally used in the simulation was of bulk material, which would be relatively higher than that of film. Therefore, to ensure the accuracy of the simulation and the guiding significance of film preparation for B_4_C/Si mirror system, it was necessary to study the effect of density parameter change. The reflectivity of different film densities was also re-calculated, and the results are shown in Table 2. In the same incident condition, the difference of reflectivity was mostly around 0.0001, which could be ignored in the simulation. With only at 15 keV, the difference was about 0.001, which was related to the atomic scattering factor calculated from photoabsorption cross section [45]. And accordingly, the data of energy deposition depth, damage threshold, film absorptivity and photoelectron emission rate are plotted in Figure 4. 

The results showed that there was no significant difference in energy deposition depth with the low incident energy at 3 keV. The main reason was that photoelectric absorption had the largest cross section. The interaction was mainly related to the incident energy, leading to the similar energy deposition depth. In addition, due to the relatively low energy, the optical path of the secondary electron was short, whose energy deposition depth was small correspondingly. With the increase in photon energy, the energy deposition depth was deeper in general. Though there was small deposition depth at 3 keV, it was also concentrated at about 60–80 nm. It has exceeded the thickness of the film (50 nm). It could mean that the energy had been deposited in the substrate. And with the increase in the photon energy, the damage length in Si substrate was deeper. It is also shown in Figure 4c. When the photon energy was above 10 keV, the film absorptivity decreased sharply. Energy would be concentrated at the interface and substrate. The damage thresholds of films with different density were consistent as shown in Figure 4b. Through the results, slight differences with film densities can be observed. The damage threshold was the largest with a density of 2.48 g/cm^3^. This was mainly due to the high energy deposition depth and reflectivity with little difference. The obvious atomic and electron behavior characteristics at this density are shown in all data in Figure 4. It could demonstrate that the density of the B_4_C film could maintain 2.48 g/cm^3^ as far as possible during the preparation for application. It enabled the mirror of B_4_C/Si to maintain higher damage characteristics and longer service life under the same conditions. Additionally, from the perspective of engineering applications, the calculated damage threshold was a relatively safe value for the system when the material density was taken as the bulk material density with 2.52 g/cm^3^. However, it is difficult to achieve this density in the preparation process. It should be emphasized that the influence of the damage threshold of the film was a complex problem. It was affected by many characteristics, such as surface roughness and film thickness. The density of the film was discussed in this work only for the condition that other properties were consistent.

### 3.3. The Influence of Film Density with the Designed Grazing Incident Angle

It is shown in Figure 3 and Figure 4b that the damage threshold decreased when the photon energy exceeded 15 keV, which is mainly related to reflectivity. It was illustrated that the incident angle should be less than 2 mrad for applications of the B_4_C/Si mirror system. Therefore, under the specific incident photon energy, selecting the appropriate grazing angle as the incident parameter within the cutoff grazing angle in practical applications is a crucial issue [46]. The designed incident parameter was related to the distance parameters of the actual optical path. The relative parameters are shown in Table 3. In order to study the damage threshold of the B_4_C/Si system with the practical application, the simulation was carried out. And to characterize the relationship between energy deposition depth, photoelectron emission rate and damage threshold, the data of each irradiation parameter were obtained through the simulation, as plotted in Figure 5. The results showed that there was a positive correlation between the damage threshold and the incident light parameters in the range of 3~25 keV. It could be seen in Figure 5a that the energy deposition depth increased with the incident parameters. When the incident parameter was 1.7 mrad@15 keV, there was a peak value among all the parameters. And then, the energy deposition depth returned to normal fluctuations. At 15 keV, the atoms in B_4_C had a large inelastic scattering cross section and photoelectron absorption cross section [37,38,39]. Therefore, the energy loss transferred to electrons, causing the secondary electron cascade process. Accordingly, the energy deposition depth and photoelectron emission rate increased. An inflection point was represented in the data image, which also showed the change in the reflectivity [40]. Therefore, these two physical parameters would have a positively relationship on the threshold damage. In addition, compared with Figure 4b, when the grazing incident angle decreased, the damage threshold increased. It was mainly related to the optical path growth of the system [46]. Thus, the properly incident parameter should be chosen in practical applications.

Thus, the distribution of enthalpy of B_4_C/Si with depth and time was obtained using a one-dimensional heat transfer model. Due to the little difference of the progress, only the results of 3 keV and 25 keV have been shown in Figure 6. The results showed that there was no heat transfer between electrons and the lattice before 0.01 ps. And the energy was mainly deposited in B_4_C film before 1 ps. The amount of energy deposition was not enough to reach the melting point of B4C film. Here, the enthalpy of melting for B_4_C and Si was 13.95 GJ/m^3^ and 3 GJ/m^3^ [47]. As time from 0.1ps to 30 ps, energy was gradually deposited from the top layer of film to the interface and then to the substrate. Most of the energy was deposited in the Si substrate, which was also shown in Figure 4c. The interface between the film and the substrate blocked the energy transfer, which reduced the energy deposited to the substrate at low energy incident parameters. When the incident energy reached 25 keV, the depth of energy deposition increased obviously. Although it also had the barrier of the interface, it was difficult to resist the energy deposition caused by the increase in incident energy. The enthalpy value reached the melting point of the substrate at the interface. Therefore, the melting at the interface caused the damage of the system. It could be understood that the photon energy had reached the K layer electron binding energy of Si, and there were a lot of photoelectrons at the interface. Only a small fraction of photoelectrons were emitted from the surface of B_4_C film, as shown in Figure 5b. Most of the photoelectrons moved into the substrate and carried a lot of energy. Finally, it reached the melting point of Si, which began from the interface firstly, as shown in Figure 6c. In addition, the depth and total amount of energy deposition increased significantly when the incident energy increased to 25 keV. Meanwhile, it was considered to have reached the damage threshold of the system under this circumstance.

To directly represent the relationship between the density of B_4_C and the damage threshold under specific incident parameters, the related data are plotted in Figure 7. Through the results, the damage thresholds in different incident energy levels were different. It could be obviously seen that damage threshold increased with the incident energy of the designed grazing angle. Due to the decrease in grazing incidence angle, compared with that of 2 mrad in Figure 4b, the damage threshold increased at the same incident energy. This occurred mainly because the free path of the secondary electron increased. Accordingly, energy deposition would be deeper. Under the same and certain incident parameters, the film density had little influence on the damage threshold. Although there was some fluctuating, it presented congruently in whole. Compared with the result of Figure 4b, it demonstrated that the change in material density and incident parameters affected the damage threshold together. It should have specific analysis according to situations. However, it showed from Figure 7 that the damage threshold was higher with the film density of 2.48 g/cm^3^, correspondingly, which was consistent with the result of Figure 4. It could also suggest that when the B_4_C film was prepared, a higher damage threshold could be obtained by keeping the density at 2.48 g/cm^3^, no matter what the incident parameter conditions. It could be significant guidance for the preparation for the film. In addition, it should be pointed out that this simulation was only set for the density change in the film. There was no doubt that the film preparation process was also affected by surface roughness and other treatment conditions, which was more complicated. It could be explored in the future work.

## 4. Conclusions

In this work, the damage threshold of B_4_C with theoretical density (2.52 g/cm^3^) was calculated by the Monte Carlo method under the incident energy of 3, 10, 15, 20, 25 keV and grazing incident angle of 2 mrad. The results showed that the damage threshold decreased sharply when the incident energy exceed 15 keV. It was mainly due to the change in reflectivity, which was related to the photoelectric absorption cross section of the constituent atoms. In addition, the energy deposition depth and damage threshold at different photon energy for B_4_C/Si with various density was studied. It showed that the energy deposition depth was positively correlated with the photon energy. At the same incident energy, the change im material density could affect the energy deposition depth, but had little effect on the damage threshold. In general, the system with B_4_C film’s density of 2.48 g/cm^3^ possessed a relatively stable and high damage threshold. It could also be seen that the damage threshold decreased sharply when the energy exceeded 15 keV at the grazing incident angle of 2 mrad. It should appropriately reduce the grazing incident angle when the energy increased. Therefore, the designed incident parameter for practical application was investigated. It showed that the energy deposition depth, photoelectron emission rate and damage threshold gradually increased with the energy. The distribution of enthalpy of B_4_C/Si with depth and time also declared that with the increase in incident energy, the energy deposited on the Si substrate gradually increased in quantity. And in the interface region there would be easily enough to reach the melting point of Si substate, causing damage to the X-ray mirror. In order to clarify the property density of B_4_C film under the specific incident parameters, it was found that damage threshold was basically consistent with different density of film. However, the density of 2.48 g/cm^3^ could guarantee the high damage threshold in general. Therefore, it was necessary to maintain the stability of the material density during preparation and processing. The uniformity of the material should be maintained for as long as possible. It will provide guidance for processing optimization of mirror film materials. It could also promote the damage threshold study of X-ray mirror film.

## Figures and Tables

**Figure 1 materials-17-01026-f001:**
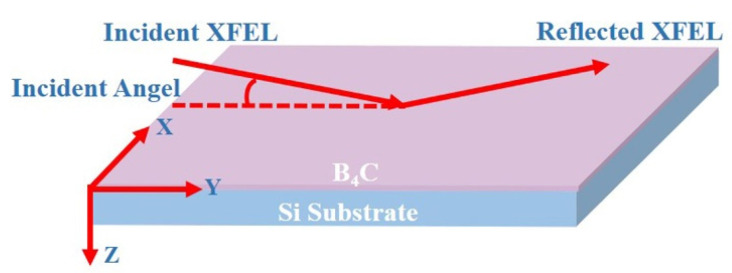
The diagram of the sample.

**Figure 2 materials-17-01026-f002:**
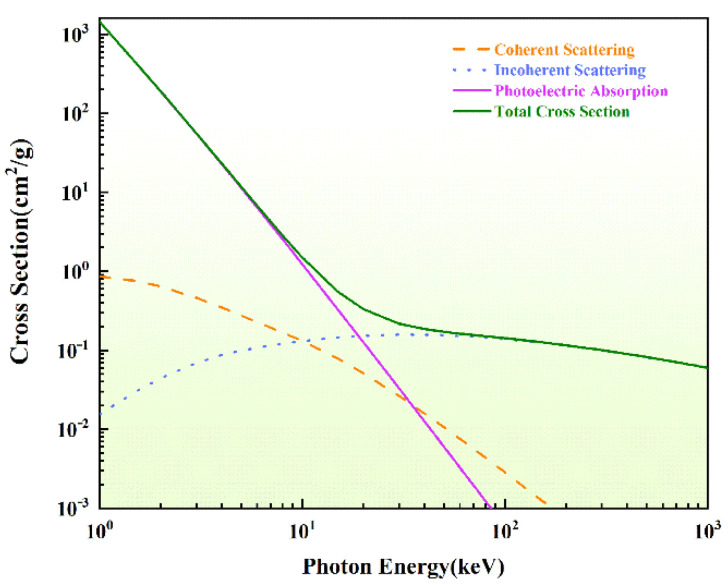
Relationship between cross section and photon energy of B4C.

**Figure 3 materials-17-01026-f003:**
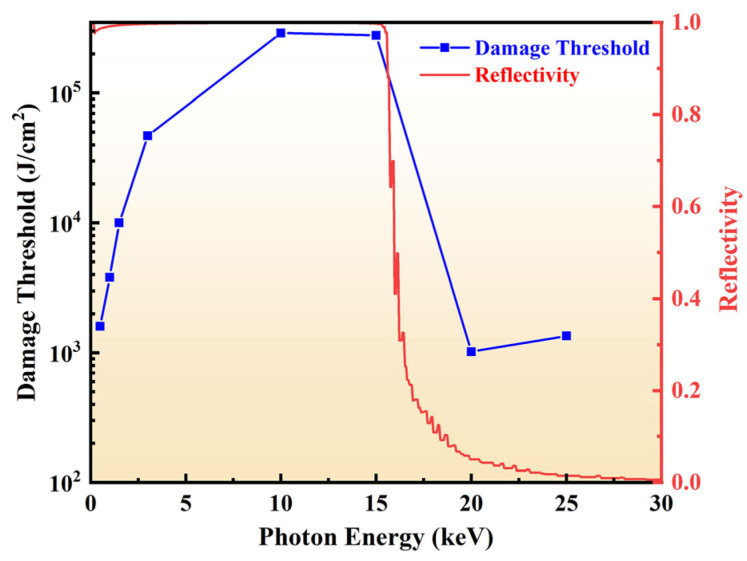
Damage threshold and reflectivity at different photon energy for B_4_C/Si.

**Figure 4 materials-17-01026-f004:**
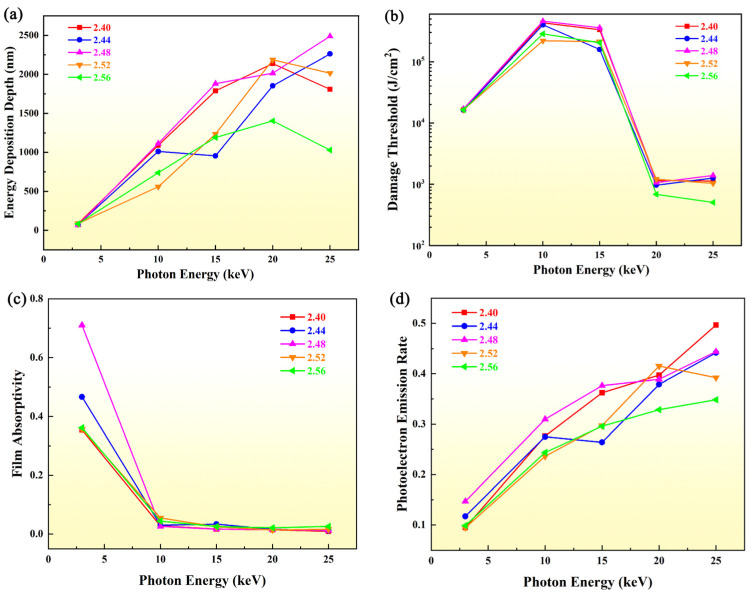
Energy deposition depth (**a**), damage threshold (**b**), film absorptivity (**c**) and photoelectron emission rate (**d**) with different photon energy at different density of B_4_C film.

**Figure 5 materials-17-01026-f005:**
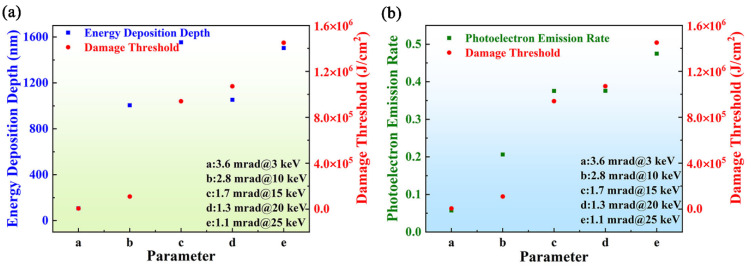
Energy deposition depth with damage threshold (**a**) and the photoelectron emission rate with damage threshold (**b**) at different incident parameter for B_4_C/Si.

**Figure 6 materials-17-01026-f006:**
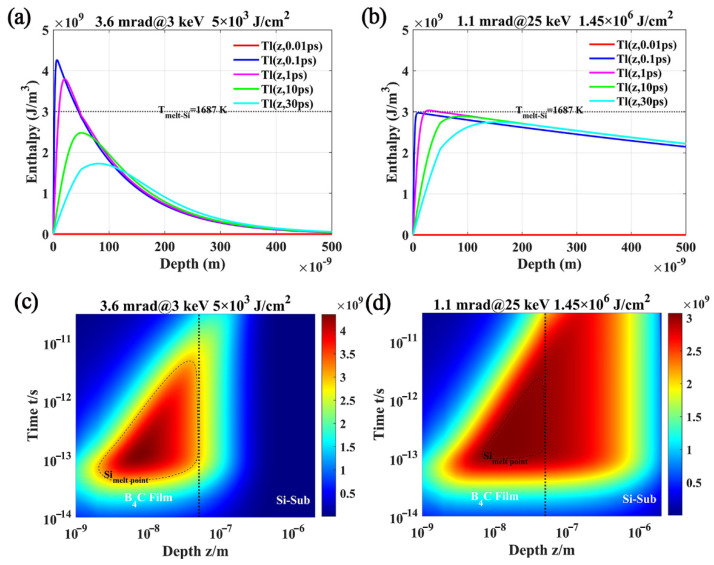
The distribution of enthalpy of B_4_C/Si with depth and time. (**a**) Enthalpy-depth distribution with 3.6 mrad@3 keV; (**b**) enthalpy-depth distribution with 1.1 mrad@25 keV; (**c**) time-depth for distribution of enthalpy with 3.6 mrad@3 keV; (**d**) time-depth for distribution of enthalpy with 1.1 mrad@25 keV.

**Figure 7 materials-17-01026-f007:**
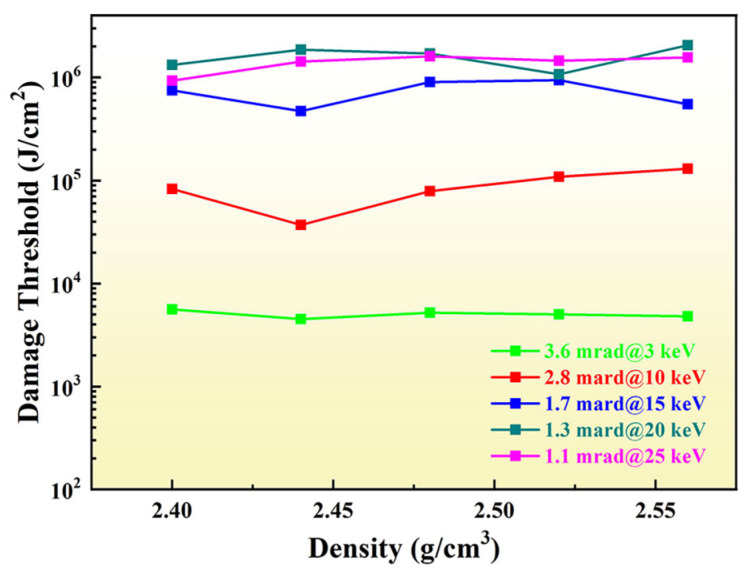
The damage threshold with the change in density of film at designed parameters.

**Table 1 materials-17-01026-t001:** The cross section derived from the simulation results compared with that in database [44]. The simulation error was ±0.01 cm^2^/g.

Photon Energy (keV)	Coherence Scattering (cm^2^/g)		Incoherence Scattering (cm^2^/g)		Photoelectric Absorption (cm^2^/g)		Total Cross Section (cm^2^/g)	
	Simulation	Database	Simulation	Database	Simulation	Database	Simulation	Database
3	0.48	0.46	0.07	0.07	55.57	55.63	56.12	56.16
10	0.13	0.13	0.13	0.13	1.231	1.24	1.49	1.50
15	0.08	0.08	0.15	0.15	0.33	0.33	0.56	0.55
20	0.05	0.05	0.15	0.15	0.13	0.13	0.33	0.33
25	0.04	0.03	0.16	0.16	0.06	0.03	0.25	0.22

**Table 2 materials-17-01026-t002:** The reflectivity of different film densities with photon energy [45].

Density of B_4_C Film (g/cm^3^)	Reflectivity with Photon Energy (2 mrad)				
	3 keV	10 keV	15 keV	20 keV	25 keV
2.40	0.9967	0.9988	0.9952	0.0498	0.0149
2.44	0.9968	0.9989	0.9967	0.0498	0.0149
2.48	0.9968	0.9989	0.9966	0.0498	0.0149
2.52	0.9968	0.9989	0.9974	0.0497	0.0149
2.56	0.9968	0.9989	0.9973	0.0497	0.0149
Standard deviations	0.00004	0.00004	0.0009	0.00005	0

**Table 3 materials-17-01026-t003:** The crucial designed incident parameters of the simulation.

Materials	B_4_C				
Photon energy (keV)	3	10	15	20	25
Grazing angle (mrad)	3.6	2.8	1.7	1.3	1.1
Reflectivity	0.9940	0.9973	0.9990	0.9992	0.9990

## Data Availability

The data presented in this study are available on request from the corresponding author (due to privacy).

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
