# Peer review of "The Influence of B4C Film Density on Damage Threshold Based on Monte Carlo Method for X-ray Mirror"

_materials, 2024, doi:10.3390/ma17051026_

Round 1

Reviewer 1 Report

Comments and Suggestions for Authors

I read the manuscript “The influence of B4C film density on damage threshold based on Monte Carlo method for X-ray mirror”. The manuscript written by T. Sui et al; described interesting theoretical parameters of B4C film to be considered for applications in beamlines (XFEL). The authors suggest from the Monte Carlo simulations that the damage threshold of B4C was noticed with a theoretical density of 2.52 g/cm3. The damage was monitored under the incident energy of 3, 10, 15, 20, 25 keV and grazing incident angle of 2 mrad. An interesting contribution is observed when the damage threshold decreases sharply when the incident energy exceeds 15 keV. The authors also recommended maintaining the density at this value when preparing the film and to ensure the uniformity and consistency of the film material. This work complements the information previously published by J. Cao et al; and Follath et al. In general, the Monte Carlo simulation and analyses seem to have been performed properly and the conclusions reached are in line with the results presented. Certainly, the theme of this manuscript is suitable for the journal and surely of wide interest to the reading community of Materials (MDPI). The references used are fine in number and actual. The only suggestion I can make to make it more amenable for reading is that standard deviations of all values given in Tables 1 and 2 should be presented. I believe that the manuscript can be accepted for publication once the authors attend to this query.

Author Response

1. Summary

Thank you very much for detailed and very helpful reports. In response to the comments, we modified the manuscript and the related tables, as shown in revised manuscript. Hope that the reviewer and editor could be satisfied with the response and modified manuscript. Please find attached the revised manuscript that we resubmit for publication in Materials.

2. Questions for General Evaluation

Reviewer’s Evaluation

Response and Revisions

Does the introduction provide sufficient background and include all relevant references?

Yes

Thank you.

Are all the cited references relevant to the research?

Yes

Thank you.

Is the research design appropriate?

Yes

Thank you.

Are the methods adequately described?

Yes

Thank you.

Are the results clearly presented?

Yes

Thank you.

Are the conclusions supported by the results?

Yes

Thank you.

3. Point-by-point response to Comments and Suggestions for Authors

Comments 1: I read the manuscript “The influence of B4C film density on damage threshold based on Monte Carlo method for X-ray mirror”. The manuscript written by T. Sui et al; described interesting theoretical parameters of B4C film to be considered for applications in beamlines (XFEL). The authors suggest from the Monte Carlo simulations that the damage threshold of B4C was noticed with a theoretical density of 2.52 g/cm3. The damage was monitored under the incident energy of 3, 10, 15, 20, 25 keV and grazing incident angle of 2 mrad. An interesting contribution is observed when the damage threshold decreases sharply when the incident energy exceeds 15 keV. The authors also recommended maintaining the density at this value when preparing the film and to ensure the uniformity and consistency of the film material. This work complements the information previously published by J. Cao et al; and Follath et al. In general, the Monte Carlo simulation and analyses seem to have been performed properly and the conclusions reached are in line with the results presented. Certainly, the theme of this manuscript is suitable for the journal and surely of wide interest to the reading community of Materials (MDPI). The references used are fine in number and actual. The only suggestion I can make to make it more amenable for reading is that standard deviations of all values given in Tables 1 and 2 should be presented. I believe that the manuscript can be accepted for publication once the authors attend to this query.

Response 1: Thank you very much for the positive and helpful comments of the manuscript. According to the referee’s suggestion, we have modified the related issues. It should be noted that since the data in Table 1 was the comparison between the simulation and database value in each photon energy with different cross section, there was no relevant standard deviation in a strict sense. However, the calculation error was added in the manuscript to make the simulation data more rigorous. For the data in Table 2, the standard deviation was given according to the reviewer’s suggestion. The related modification was shown in Table 1 and 2 with highlight, which could be found in page 5 line 197 and page 7 line 261. Hope that the reviewer and editor could be satisfied with the modification.

Reviewer 2 Report

Comments and Suggestions for Authors

The authors report a computational study,based on MC methods, inorder to study
the damage threshold of a B4C film.
The topic is interest and certianly falls within the interest of readership.
The whole presentation is satisfactory.
My my main (and serious concerns) relies on the reliability of the applied methodology.
Not many details are given the simulation details section.
In such processes the underlying physics is complicated, due to various phenomena involve.
The analysis provided in section 3.1 is not fully convincible.
I suggest the authors to validate MC method with similar applied in the literature and used for studying such
processes.

Author Response

  1. Summary

Thank you for the helpful comments regarding to our manuscript. According to the suggestions, we modified the manuscript as shown in revised version. Hope that the reviewer and editor could be satisfied with the response and modified manuscript. Please find attached the revised manuscript that we resubmit for publication in Materials.

2. Questions for General Evaluation

Reviewer’s Evaluation

Response and Revisions

Does the introduction provide sufficient background and include all relevant references?

Yes

Thank you.

Are all the cited references relevant to the research?

Yes

Thank you.

Is the research design appropriate?

Yes

Thank you.

Are the methods adequately described?

Can be improved

Thank you for your suggestions. The details were shown in Part 3.

Are the results clearly presented?

Can be improved

Thank you for your suggestions. The details were shown in Part 3.

Are the conclusions supported by the results?

Can be improved

Thank you for your suggestions. The details were shown in Part 3.

  1. Point-by-point response to Comments and Suggestions for Authors

Comments 1: The authors report a computational study, based on MC methods, in order to study the damage threshold of a B4C film. The topic is interest and certainly falls within the interest of readership. The whole presentation is satisfactory. My main (and serious concerns) relies on the reliability of the applied methodology. Not many details are given the simulation details section. In such processes the underlying physics is complicated, due to various phenomena involve. The analysis provided in section 3.1 is not fully convincible. I suggest the authors to validate MC method with similar applied in the literature and used for studying such processes.

Response 1: Thank you for the positive comments and very helpful suggestions of our manuscript. Firstly, according to the main and serious concerns of reviewer, it could be regarded as the reliability of the simulation, which mostly relied on the Monte Carlo method. Monte Carlo method is based on statistical sampling theory. Random numbers are used for sampling experiment or random simulation. It is a numerical method for solving approximate solutions of engineering technical problems. And it can be applied to random function with arbitrary distribution, solving both uncertain and definite problems. The method has been applied to the research of the interaction between photon and matter, such as Ref.15,16,42,44 (Appl. Surf. Sci., 2020, 501, 146952.; Opt. Express., 2018, 26(15):19665-19685.; Chinese Opt. Lett., 2023, 21(2):023401.; Proc. SPIE, 2011, 4500, 51.) etc. Therefore, the reliability could be guaranteed. Secondly, the simulation in this work was implemented by Geant4 software based on Monte Carlo method. It is an open-source software with specific handbook, and some details were published in Ref.33-35 (IEEE T. Nucl. Sci., 2006, 53(1):270-278.; Nucl. Instrum. Meth. A., 2003, 506(3):250-303.; Nucl. Instrum. Meth. A., 2016, 835, 186-225.). All of the simulation were completed in Geant4, including light source and substance. The spatial distribution of the photons was Gaussian distribution with the number of 105. And the area of the irradiation of B4C was 7.27×10-7~1.07×10-6 m2 in order to be consistent with the practical application value. The photoelectric effect, Auger effect and secondary electron cascade would occur according to the cross-section data which would be extracted from the database. Thus, all of the simulation details were shown in part 2.2. Finally, just as the reviewer suggested, similar work published in references and used for studying such processes have also been shown in appropriate position in manuscript, such as Ref. 42-45,47(Chinese Opt. Lett., 2023, 21(2):023401.; Conceptual design report: X-ray optics and beam transport, Published, 2011, Physics, Engineering; Proc. SPIE, 2011, 4500, 51.; Proc. SPIE,2007, 6586, 658605.; Atomic Data and Nuclear Data Tables. 1993, 54 (2), 181-342.), to prove the reliability of the study. The results showed that it was consistent with that of published work, Ref. 42 (Chinese Opt. Lett., 2023, 21(2):023401.)

Above all, we hope that the reviewer and editor could be satisfied with the response and modified manuscript.

Reviewer 3 Report

Comments and Suggestions for Authors

This manuscript describes a monte carlo simulation studies on the damage threshold of B4C film vs its density. This work is of significance to the research community. The methods are clearly described. However, I would recommend major revision as I spotted several issues:
1. The authors need to rewrite the abstract as some sentences are too hard to read.

2. The authors need to rewrite section 3.1 as many claims there are simply not correct. For example, "electron couple effect" should be pair production instead. "When the energy of 166 the incident photon is less than 1 MeV, it is mainly the photoelectric effect." is simply not true as the probability is also a function of proton number. 

3. According to equation (3), the damage threshold is a linear function of the density and reflectivity. The simulation results seem to match this but the authors need to derive a mapping. 

Comments on the Quality of English Language

The authors need to work on some sentences. For example, "It is taken boron carbide 11 film and monocrystalline silicon substrate as example, the influence of density on the damage 12 threshold was studied by Monte Carlo and heat-conduction methods." I have no idea what you mean here. 

Author Response

  1. Summary

Thank you very much for the detailed and helpful suggestions regarding to our manuscript. In response to the reports, we modified the manuscript as shown in revised version. Hope that the reviewer and editor could be satisfied with the response and modified manuscript. Please find attached the revised manuscript that we resubmit for publication in Materials.

2. Questions for General Evaluation

Reviewer’s Evaluation

Response and Revisions

Does the introduction provide sufficient background and include all relevant references?

Can be improved

Thank you and the details was shown in Part3.

Are all the cited references relevant to the research?

Yes

Thank you.

Is the research design appropriate?

Yes

Thank you.

Are the methods adequately described?

Yes

Thank you.

Are the results clearly presented?

Can be improved

Thank you and the details was shown in Part 3.

Are the conclusions supported by the results?

Must be improved

Thank you and the details was shown in Part 3.

  1. Point-by-point response to Comments and Suggestions for Authors

Comments 1: This manuscript describes Monte Carlo simulation studies on the damage threshold of B4C film vs its density. This work is of significance to the research community. The methods are clearly described. However, I would recommend major revision as I spotted several issues: 1. The authors need to rewrite the abstract as some sentences are too hard to read.

Response 1: Thank you for your suggestion and sorry for the expression. According to the referee’s suggestion, we have modified the abstract as shown in Page 1 line 8-13. The revised abstract are as follows:

Abstract: The uniformity and consistency of X-ray mirror film materials prepared by experiment methods are difficult to guarantee completely. It will directly affect the service life of free electron laser devices in addition to its own optical properties. Therefore, the quality of the film material, especially the density, has a critical effect on its application. Boron carbide film and monocrystal-line silicon substrate were suitable examples to explore their influence of density on the damage threshold based on Monte Carlo and heat-conduction methods. Through simulation results, it was found that the change of film density could affect the energy deposition depth and damage threshold. When the film density was 2.48 g/cm3, it had relatively high damage threshold in all energy ranges. And then the specific incident parameter for practical application was investigated. It was found that the damage mechanism of the B4C/Si was the melting of the interface. And the damage threshold was also higher with the film density of 2.48 g/cm3. Therefore, it was recommended to maintain the density at this value as far as possible when preparing the film, and to ensure the uniformity and consistency of the film material.

Comments 2: The authors need to rewrite section 3.1 as many claims there are simply not correct. For example, "electron couple effect" should be pair production instead. "When the energy of 166 the incident photon is less than 1 MeV, it is mainly the photoelectric effect." is simply not true as the probability is also a function of proton number.

Response 2: Thank you for your suggestion and sorry for the inappropriate expression. The related sentences have been modified as shown in Page 4 line 165-176. It was modified as “Generally, there are three main interactions between photons and matter, including photoelectric effect, Compton scattering and pair production. The photoelectric effect is dominant for low-energy photons and substances with high atomic number.”

Comments 3: According to equation (3), the damage threshold is a linear function of the density and reflectivity. The simulation results seem to match this but the authors need to derive a mapping.

Response 3: Thank you for the comment. According to equation (3), the damage threshold is indeed a function of the density and reflectivity. However, it also includes another parameter d, representing the energy deposition depth. It was also related to the density and reflectivity based on the simulation of Monte Carlo method, as shown in Figure 4 (a). When the density of system changes, the reflectivity and energy deposition depth would change, the relationship between them is not a simple linear relationship. Especially, incident energy also plays a significant role, which was proved by the References 44,45. Thus, it could not derive a simple mapping with complex relationship. However, the simulation results showed that they were matched each other, proving the reliability of the results.

4. Response to Comments on the Quality of English Language

Point 1: The authors need to work on some sentences. For example, "It is taken boron carbide 11 film and monocrystalline silicon substrate as example, the influence of density on the damage 12 threshold was studied by Monte Carlo and heat-conduction methods." I have no idea what you mean here.

Response 1: Thank you for your suggestion and sorry for some obscure sentence. It was modified as “Boron carbide film and monocrystalline silicon substrate were suitable examples to explore their influence of density on the damage threshold based on Monte Carlo and heat-conduction methods. “, which was shown in Page 1 line 11-13.

Round 2

Reviewer 2 Report

Comments and Suggestions for Authors

The authors have revised substantially their research.

Reviewer 3 Report

Comments and Suggestions for Authors

Thank you for addressing my concerns. I recommend its acceptance.